# OpenReview forum: "Optimizing Preference Alignment with Differentiable NDCG Ranking"
_ICLR.cc/2025/Conference — Submitted to ICLR 2025_

### Official Review · Reviewer_mWpe · 2024-10-28

**Soundness:** 3
**Presentation:** 2
**Contribution:** 2
**Rating:** 6
**Confidence:** 4

**Summary:**

The paper introduces a novel method to improve the alignment of large language models (LLMs) with human preferences. This method, called Direct Ranking Preference Optimization (DRPO), views human preference alignment as a Learning-to-Rank (LTR) task and employs the Normalized Discounted Cumulative Gain (NDCG) metric for direct optimization from ranked response lists.

Contributions of the paper include:

**Novel Method**: DRPO treats human preference alignment as a listwise ranking problem, using ranking preference data to optimize LLMs.

**Adaptive Rank Policy Score**: A new ranking score computation strategy is introduced to dynamically adjust score margins based on relative positions.

**Differentiable NDCG (diffNDCG)**: A differentiable approximation of the NDCG metric is developed to serve as the loss function, allowing for more accurate alignment of LLM responses with human preferences.

**Strengths:**

1. The performance gain from the proposed method appears to be quite significant.
2. The code and models are publicly available for review.
3. Some revise has been incorporated. I’ve reviewed the paper previously for a different venue, where I’ve raised questions on the choice of discount factors and comparison with other differentiable approximation methods of NDCG. Both have been incorporated in the main text as ablation studies.

**Weaknesses:**

1. However, some other discussion in the previous review has been ignored. For example, in the previous review, I’ve asked for a novelty discussion against LiPO work. The discussion is still missing. The authors only mention “which (diffNDCG) can utilize list data more effectively than existing methods” in Line 812 and no discussion like “reward hacking challenge” is mentioned. Is that because diffNDCG is not a proper solution to the challenge as well? Or due to some other reasons, which are unclear to me.
2. The main issue of the work is that the proposed method appears very significantly better than other differentiable ranking approaches. This is extraordinary, but hard for me as a reader why it’s so much more effective than others. Because of better differentiable quality (what in specific)? Or because the experiments were done on some specific settings? Like Qwen 0.5B and 1.8B. Both sizes are a bit too small for functioning Large Language Models. For example, the results on Mistral-7B are less significant.
3. The results are a bit too dense. Very tiny fonts, hard to read and follow.

**Questions:**

1. In Figure 3, it appears that LiPO grows with list size a bit faster than DRPO, will they at some point cross in terms of RM win rate, or the gap will keep eventually?

---

> ### Author Response · Authors · 2024-11-25
>
> **1. Lack novelty discussion against LiPO work.**
>
> LiPO [1] leverages the LambdaRank loss [2] to learn human preferences from ranking preference data. This loss function integrates DCG weighting, which provides listwise-aware importance factors, into the pairwise logistic ranking loss [3]. LiPO uses this loss as an indirect way to optimize NDCG for ranked lists, where [4] shows that LambdaRank loss optimizes a coarse upper bound of NDCG. In contrast, we implement differentiable sorting networks to sort responses and obtain differentiable permutation matrices. By leveraging these matrices, we develop a differentiable NDCG metric (diffNDCG), enabling direct optimization of NDCG for ranked lists.
>
> Additionally, we develop the Adaptive Rank Policy Score, a novel ranking strategy that maximizes the absolute likelihood of preferred responses while dynamically adjusting score margins between preferred and non-preferred responses based on their relative positions in the ranked list. In future work, we will investigate whether this approach can help mitigate the reward hacking challenge.
>
> Furthermore, we investigate the relationship between NDCG and WinRate metrics in LLM evaluation. Our analysis demonstrates that NDCG exhibits a stronger correlation with WinRate compared to other targets used in existing methods[5], suggesting its potential as both an evaluation metric and optimization objective.
>
> **2. The justification for our DRPO being significantly better than other differentiable ranking approaches.**
>
> In our experiments, we compared several differentiable sorting methods, including Neural Sort [6], Differentiable Sorting Network [7], and Fast Soft Sort [8], and other classic LTR loss. All methods were implemented following their official repositories with default parameter settings. Both our DRPO and LiPO significantly outperform Neural Sort and Differentiable Sorting Network, while showing modest improvements over Fast Soft Sort.
>
> We hypothesize that the performance gap stems from both Differentiable Sorting Network and Neural Sort using cross-entropy to learn the mapping between current and optimal permutation matrices. Given that LLM output scores typically exhibit small variations, this approach may make it challenging to learn effective permutation matrices.
>
> In contrast, Fast Sort approaches the problem by casting differentiable sorting and ranking as projections onto the permutahedron (the convex hull of all permutations) and employing isotonic optimization. This approach naturally maintains certain margins between different items in the list, which facilitates the learning of LLM preference alignment.
>
> In Mistral Experiments, we compared DRPO against recent competitive methods on the UltraFeedback dataset, including LiPO. Due to the effectiveness of our DRPO, we still achieved modest performance gains over these strong baselines.
>
> **3. The results are a bit too dense.**
>
> In our future revision, we will improve the tables' readability. We have also noted that our font sizes are comparable to or larger than those in several ICLR accepted papers such as [9,10,11].
>
> **4. Comparative Analysis of DRPO and LiPO with Larger List Sizes**
>
> Given that our method achieves strong performance even with smaller list sizes (K=2, K=4), the potential for improvement at larger list sizes is relatively limited. This explains why LiPO appears to demonstrate a faster growth rate than DRPO as the list size increases. Based on this trend and our experimental observations, we expect the performance gap between DRPO and LiPO to maintain, rather than cross, as DRPO has already approached near-optimal performance at this list sizes.
>
> To verify performance with larger lists, we extended our dataset to K=12 by generating four additional responses per prompt using Qwen1.5-1.8B-SFT. Results below confirm DRPO maintains its advantage over LiPO at larger list sizes.
>
> |Method|RM Winrate(vs Chosen)|RM Winrate(vs SFT)|
> |---|---|---|
> |LiPO(K=12)    | 52.92%($\pm2.99$) | 74.21%($\pm2.76$) |
> |DRPO(K=12)    |61.91%($\pm2.16$)|81.44%($\pm2.02$)|
>
> [1] LiPO: Listwise Preference Optimization through Learning-to-Rank, arXiv:2402.01878.
> [2] Learning to Rank with Nonsmooth Cost Functions, Neurips 2006.
> [3] Learning to rank using gradient descent, ICML2005.
> [4] The LambdaLoss Framework for Ranking Metric Optimization, CIKM 2018.
> [5] Preference Learning Algorithms Do Not Learn Preference Rankings, Neurips 2024.
> [6] Stochastic Optimization of Sorting Networks via Continuous Relaxations. ICLR 2019.
> [7] Differentiable Sorting Networks for Scalable Sorting and Ranking Supervision. ICML 2021.
> [8] Fast Differentiable Sorting and Ranking, ICML 2020.
> [9] MathVista: Evaluating Mathematical Reasoning of Foundation Models in Visual Contexts, ICLR 2024.
> [10] Enhancing Neural Subset Selection: Integrating Background Information into Set Representations, ICLR 2024.
> [11] Test-time Adaptation against Multi-modal Reliability Bias, ICLR 2024.

---

> > ### Author Response · Authors · 2024-11-25
> >
> > Dear Reviewer mWpe,
> >
> > Your profound insights have been tremendously valuable in guiding our work. Understanding your demanding schedule, we deeply appreciate your dedication to reviewing our paper. We have invested considerable effort in addressing the concerns raised during the initial review phase, though we have yet to receive feedback on these revisions.
> >
> > Given that the rebuttal period is drawing to a close, we wanted to respectfully follow up. Your insights on our responses would be invaluable to us at this stage. Should there be any remaining concerns or areas needing further refinement, we stand ready to make additional improvements to our work.
> >
> > We sincerely thank you for your time and constructive feedback throughout this process.

---

> > ### Comment · Reviewer_mWpe · 2024-11-25
> >
> > Thanks for clarifications on my questions. Conditioned on the additional results and discussions to be added, I'll raise my overall rating.

---

> > ### Comment · Reviewer_nU6a · 2024-11-25
> >
> > > In contrast, we implement differentiable sorting networks to sort responses and obtain differentiable permutation matrices. By leveraging these matrices, we develop a differentiable NDCG metric (diffNDCG), enabling direct optimization of NDCG for ranked lists.
> >
> > This statement is incorrect.You find solutions as a **doubly stochastic** permutation matrix which is different from a permutation matrix (with 0/1 entries). As a result, the proposed approach is no more direct than using Lambda Rank in the current context.

---

> ### Author Response · Authors · 2024-11-26
>
> We appreciate your careful observations. Due to the non-differentiable nature of NDCG, any optimization approach must employ some form of approximation. Our method directly optimizes the ranking metric through its differentiable approximation, rather than optimizing other objectives.

---

### Official Review · Reviewer_nU6a · 2024-11-02

**Soundness:** 2
**Presentation:** 2
**Contribution:** 2
**Rating:** 3
**Confidence:** 4

**Summary:**

The paper proposes a DRPO method, a modification of DPO, for aligning Large Language Models (LLMs) with human preferences. The core idea of the paper is to leverage the NDCG metric and corresponding continuous proxies to augment the DPO approach, which relies solely on pairwise preference. The authors introduce the "Adaptive Rank Policy Score" (Equation (5)), an approach that assigns each candidate (LLM response) a score that combines the log-likelihood of the response (according to the current policy) with position factors. The policy is then updated by comparing the generated order with the ground truth ranking via a differentiable NDCG proxy, diffNDCG. My vote is a "reject" due to issues in the evaluation methodology and the lack of a rigorous analysis of the proposed scoring scheme. I found the empirical evaluations to be limited, with no insights on the assumptions under which the proposed approach might generalize to a broader setting. I believe the paper is not ready for presentation at ICLR.

**Strengths:**

The idea of using the log likelihoods as the gains in the NDCG metrics is interesting, and might have an impact.

**Weaknesses:**

In the paragraph on line 061, the authors argue that "win rate" is more correlated with NDCG metrics. However, the evidence they provide in Figure 4 and Table 11 is limited. The main issue with Figure 4 is that the numbers are conditioned on the use of the proposed DRPO method. Thus, Figure 4 does not provide information on what might happen with a different method. From this perspective, Figure 4 should be removed from the paper. In Table 11, the authors calculate Pearson's correlation between NDCG and "win rate"; however, the table lacks the correlation with the pairwise approach used in the original DPO. Table 11 does not support differentiation between the pairwise approach and NDCG ranking. Furthermore, the proper way to compare correlation between ranked lists is the Kendall Tau distance.

I somewhat disagree with the statement on line 096. A well-established loss from the LTR community is LambdaRank, which relies on pairwise comparisons and builds a preference model somewhat similar to DPO. Given this connection, the original DPO paper optimizes for NDCG metrics via pairwise preferences. While it might not be an ideal proxy (as it does not use the lambda weights), improving pairwise alignment does lead to better ranking.

On line 102, the authors discuss the contribution of "a novel differentiable NDCG" metric. This contribution is incremental, as it is a straightforward application of sorting networks (Petersen et al. 2021) to the optimization of NDCG metrics.

The main contribution of the paper is the scoring approach in Equation (5). The moving average of scores for a specific position relies on assumptions that the authors did not discuss. For example, I expect the ARPS scoring scheme to perform poorly if the distribution of examples changes drastically from one instance to another, as might be expected when aligning over different types of tasks or questions.
Table 1 contains imprecision that makes it challenging to assess the results. The entries for ListNet, PiRank, and Neural Sort are actually the PPR and ARP methods with these methods as subroutines (see line 406). However, is it PPR or ARP? Line 406 states both ("and"). Additionally, I find it misleading to use the names of these subroutines in the context of PPR and ARP; please be explicit about this distinction.

One of the main challenges in assessing the quality of the proposed methods is that the ground truth is obtained via other models. The authors acknowledge this in Appendix B, but I suggest moving this acknowledgment to the beginning of the paper to provide clarity upfront about what the reader can expect. As it stands, the proposed methodology might capture biases in the evaluation models. If one switches the evaluation models, the numbers could change unpredictably.

I would like to point out that the choice of alpha in Equation (8) is critical. If the authors make comparisons with PiRank, they need to specify the temperature used in the experiment. The alpha in diffNDCG relates to the temperature in PiRank, and adjusting it controls the trade-off between convergence speed and precision of the final estimates.

**Questions:**

I encourage the authors to be more precise in their mathematical notation. In equation (5) the quantity $V_{q(y)}$ appears on both sides of the equation, which would reduce the quantity to a linear combination of $\log\pi$. Please add iteration indexing.

---

> ### Author Response · Authors · 2024-11-25
>
> In our next revision, we will explicitly clarify the distinction of score functions used, improve overall readability, and move the limitations section to the beginning of the paper for better clarity.
>
> Below are our detailed responses to the specific concerns:
>
> **1. More comparison of the correlation between NDCG and Winrate**
>
>
> Following [1], we collected checkpoints at different training steps of Qwen1.5-0.5B and Qwen1.5-1.8B on our ranking preference dataset to analyze both NDCG and ranking accuracy in relation to win rate. As shown in Figure 4, we explicitly compare these two metrics, whereas [1] only examined ranking accuracy against win rate. In Table 11, we analyze the correlations during the training process using Pearson's correlation coefficient. Specifically, we examine the correlation between NDCG changes and win rate changes, as well as between ranking accuracy changes and win rate changes. The Kendall Tau rank distance, which quantifies the ordinal association between two rankings, is not applicable for this correlation analysis between evaluation metrics.
>
> Additionally, in the revised paper, we will provide visualization plots illustrating the relationships of NDCG and ranking accuracy with win rate during the training process of different methods.
>
> **2. Clarification of Our Contributions and Relationship to Prior Work**
>
>
> Unlike traditional learning-to-rank tasks, our primary goal is to align large language models with human preferences. Although prior work like LiPO [2] has framed preference alignment as a learning-to-rank problem and leveraged LambdaRank loss [3] to learn human preferences from ranking data, limited research has focused on advancing list preference alignment. LiPO integrates DCG weighting, which provides listwise-aware importance factors, into the pairwise logistic ranking loss. LiPO uses this loss as an indirect way to optimize NDCG for ranked lists.
>
> In contrast, we implement differentiable sorting networks to sort responses based our novel score function and obtain differentiable permutation matrices. By leveraging these matrices, we develop a differentiable NDCG metric (diffNDCG), enabling direct optimization of NDCG for ranking preference data. Extensive experimental results demonstrate both the superiority of this direct optimization approach and DRPO's enhanced performance over existing baseline methods in generating high-quality responses.
>
> This approach represents a novel contribution to preference alignment tasks, as no prior work has attempted to perform human preference alignment in this direct manner. Additionally, our investigation into the relationship between NDCG and WinRate metrics in LLM evaluation reveals that NDCG demonstrates stronger correlation with WinRate compared to targets used in existing methods [1]. This correlation suggests NDCG's dual potential as both an evaluation metric and optimization objective.
>
> **3. More Explaination for our scoring approach**
>
> In our implementation of Equation 5, we clip the updated scores to ensure their interpolation does not exceed a threshold. This prevents the distribution of examples from changing drastically between instances. We will update the table representation to clearly indicate which score function is used in each case. Specifically, all Learning-to-Rank losses compared in Table 1 utilize the PPR score function. In Table 2, we then evaluate the performance changes when replacing PPR with ARP.
>
> **4. More Details: PiRank Comparison and Alpha Ablation in Eq.8**
>
> In the experimental setup, we used the implementation from the official repository of PiRank with a temperature of 1. For our method, we set alpha in Eq. 8 to 1. These hyperparameter configurations will be specified in detail in the revised paper.
>
> We conducted additional experiments to compare model performance under different values of alpha in Eq. 8. The results, shown below, indicate that varying alpha has minimal impact on model performance, and the choice of alpha = 1 serves as a balanced compromise.
>
> |Method|RM Winrate(vs Chosen)|RM Winrate(vs SFT)|
> |---|---|---|
> |DRPO($\alpha=0.1$)    |58.39%($\pm1.77$)|76.36%($\pm2.94$)|
> |DRPO($\alpha=1$)    |58.98%($\pm4.95$)|78.90%($\pm3.70$)|
> |DRPO($\alpha=10$)    |59.76%($\pm3.68$)|79.10%($\pm4.95$)|
> |DRPO($\alpha=50$)    |58.59%($\pm1.46$)|78.71%($\pm0.64$)|
>
>
>
> **5. Explanation of Notation Used in Equation 5**
>
> To enhance mathematical precision, we revise Eq.5 as follows:
> $$
> \gamma(y) = \tau \cdot q(y) - \beta \cdot {V}_{q(y)},~
> \text{where} ~ {V}_{q(y)} \gets \theta \cdot {V}_{q(y)} + (1-\theta) \cdot \frac{1}{|y|} \log \pi_\theta(y \mid x).
> $$
>
> [1] Preference Learning Algorithms Do Not Learn Preference Rankings, Neurips 2024.
> [2] LiPO: Listwise Preference Optimization through Learning-to-Rank, arXiv:2402.01878.
> [3] Learning to Rank with Nonsmooth Cost Functions, Neurips 2006.

---

> > ### Comment · Reviewer_nU6a · 2024-11-25
> >
> > Dear Authors, I urge you to update the manuscript rather than promising next revisions.
> >
> > Meanwhile, I have few more questions:
> > a) In the statement below, what is the **list** preference alignment?
> > > Although prior work like LiPO [2] has framed preference alignment as a learning-to-rank problem and leveraged LambdaRank loss [3] to learn human preferences from ranking data, limited research has focused on advancing list preference alignment.
> >
> > I strongly suspect, if one to write the definition diligently, then they would arrive at a similar framing of preference alignment via learning-to-rank, where you use a constant position discount. One of the main point of the manuscript is alignment via NDCG metrics -- that is the core metric in the learning-to-rank domain.
> >
> > b) On a comparison with PiRank and other methods. Thank you for adding results by varying alpha in your algorithm. However, Is it a fair comparison with other methods, like PiRank, where you did not tuner parameters of the corresponding methods?  Will results still hold if competing methods are tuned?
> >
> > c) In the current manuscript, there is a lack of clarity on the process of measuring winrates, and computing NDCGs. For example, to compute winrate, it is enough to have a pairwise comparison (say between to competing LLMs), yet NDCG measure the fidelity of listwise matches between human preferences and an LLM's outputs. How long are the lists? How many rounds in each measurement? What happens when the lists are of size 2? Etc.

---

> > > ### Author Response · Authors · 2024-11-27
> > >
> > > We have invested considerable effort in addressing the concerns raised during the initial review phase. In our revised manuscript, we provide detailed experimental analysis of our scoring function and comprehensive descriptions of the experimental setup. We sincerely thank you for your time and constructive feedback throughout this process and look forward to your response.

---

> ### Author Response · Authors · 2024-11-25
>
> Dear Reviewer nU6a
>
> We appreciate your feedback and have revised the manuscript accordingly. We have clarified the distinction between score functions used in learning-to-rank methods and move limitation has been relocated to preceding sections for better clarity. Furthermore, we have included comprehensive experimental details in Appendix G, including the computation methods for NDCG, Accuracy, and Win Rate metrics.
>
> As for the remaining question, we provide our response below.
>
> 1. what is list preference alignment and our contribution
>
> Aligning large language models with human preferences enhances interaction quality and safety by ensuring outputs better reflect human values. This alignment process relies on datasets of human preferences, where human evaluators or advanced AI systems (like GPT-4-turbo) rank multiple responses from the same model based on quality and appropriateness to determine which outputs better match human expectations.
>
> Current LLM alignment approaches operate through two main paradigms: The first trains a reward model using pairwise preference data, then employs RLHF methods like PPO to optimize for higher rewards. The second, exemplified by DPO, learns directly from preference pairs, eliminating the intermediate reward modeling step.
>
> Nevertheless, human preferences are not solely expressed as simple binary comparisons of preferred versus non-preferred responses; they also manifest as ranking information, an aspect that has rarely been explored in previous research.
> Therefore, **list preference alignment** represents an approach that aligns LLMs using ranked preference data rather than simple binary comparisons.
>
> Existing approaches have explored ranked preferences in different ways: RRHF[1] uses reward-model-labeled list data to optimize a pairwise contrastive objective, while other methods employ list MLE loss[2] or lambdaRank[3] loss to align LLMs with ranked preferences.
>
> In contrast, we implement differentiable sorting networks to sort responses based our novel score function and obtain differentiable permutation matrices and develop a differentiable NDCG metric (diffNDCG), enabling direct optimization of NDCG for ranking preference data. This approach represents a novel contribution to preference alignment tasks, as no prior work has attempted to perform human preference alignment in this direct manner. Additionally, our investigation into the relationship between NDCG and WinRate metrics in LLM evaluation reveals that NDCG demonstrates stronger correlation with WinRate compared to targets used in existing methods [4]. This correlation suggests NDCG's dual potential as both an evaluation metric and optimization objective.
>
>
> 2. More comparisons about Pirank
>
> We appreciate your concern about parameter tuning. We conducted a fair comparison by setting the temperature parameter to 1 for both our method and PiRank. Through extensive experimentation, we found that both Neural Sort and our odd-even differentiable sorting network demonstrate low sensitivity to temperature variations in the human preference alignment task, showing minimal performance fluctuations across different temperature settings, as evidenced by our experiments with the stepness hyperparameter $\alpha$. Since PiRank is built upon Neural Sort [5] and derives its sorting matrix from it, varying the temperature in PiRank would not lead to significant performance improvements.
>
> To further validate our findings, we experimented with PiRank across various temperature settings. The experimental results show:
>
> - Pirank With PRR Score.
>
> |Method|RM Winrate(vs Chosen)|RM Winrate(vs SFT)|
> |---|---|---|
> |Pirank($\tau=0.5$)    |35.54%($\pm3.05$)|61.71%($\pm2.27$)|
> |Pirank($\tau=1$)    |38.87%($\pm2.73$)|62.50%($\pm3.07$)|
> |Pirank($\tau=5$)    |38.28%($\pm1.99$)|61.71%($\pm2.27$)|
>
> Additional results will be provided in subsequent updates.
>
> 3. clarity on the process of measuring winrates, and computing NDCGs
>
> We have presented comprehensive details in Appendix G.3 of our revised manuscript.
>
> [1] RRHF: Rank Responses to Align Language Models with Human Feedback without tears, Neurips 2023.
>
> [2] Preference Ranking Optimization for Human Alignment, AAAI,2024.
>
> [3] LiPO: Listwise Preference Optimization through Learning-to-Rank, Arxiv:2304.05302.
>
> [4] Preference Learning Algorithms Do Not Learn Preference Rankings, Neurips 2024.
>
> [5] Stochastic Optimization of Sorting Networks via Continuous Relaxations, ICLR 2019.

---

> > ### Comment · Reviewer_nU6a · 2024-11-25
> >
> > I would like to surface two statements made by the Authors
> >
> > > Although prior work like LiPO [2] has framed preference alignment as a **learning-to-rank problem** and leveraged LambdaRank loss [3] to learn human preferences from ranking data, limited research has focused on advancing **list preference** alignment.
> >
> > and then
> >
> > > Therefore, **list preference** alignment represents an approach that aligns LLMs using **ranked preference** data rather than simple binary comparisons.
> >
> > The relation between list preference alignment and alignment via a learning-to-rank problem is contradictory from the above statments.
> >
> > In the similar note,  the statement
> > >To our best knowledge, this paper is a pioneer exploration to align LLMs with ranking human preference data.
> >
> > is incorrect given presence of LiPO work.

---

> > > ### Author Response · Authors · 2024-11-30
> > >
> > > We deeply appreciate the time and effort you have invested in reviewing our work. We understand the many demands on your schedule and are truly grateful for your thoughtful consideration of our response. Fortunately, ICLR has granted us an additional week to engage in further discussions. Your insights are invaluable to us, and we would be honored to receive any additional feedback you might provide.

---

> > > > ### Comment · Reviewer_nU6a · 2024-12-01
> > > >
> > > > I appreciate the authors' efforts in addressing some of the feedback. The additions, including more detailed hyperparameter descriptions and notation improvements, are noted and helpful. However, I must maintain my overall score of 3. In my view, the revised manuscript still does not sufficiently present the material, particularly in the context of prior work, such as LiPO.
> > > >
> > > > While diffNDCG represents an incremental improvement by applying sorting networks to optimize the NDCG metric, it does not constitute a pioneering contribution as suggested by the authors. The method is primarily an empirical improvement, serving as a better proxy for NDCG in a specific context, but it provides limited new insights or learning. Furthermore, the paper lacks theoretical guarantees to support the claims regarding diffNDCG’s superiority as a proxy for NDCG, especially given the narrow scope of the study centered on the Adaptive Rank Policy Score.
> > > >
> > > > In my opinion, the primary novelty of the work lies in Equation (5). However, the manuscript does not provide sufficient insights into the Adaptive Rank Policy Score, which appears to be yet another heuristic based on the information presented.

---

> ### Author Response · Authors · 2024-11-26
>
> We thank Reviewer nU6a for this important observation. However, the term 'pioneer' extends beyond mere chronological primacy to encompass groundbreaking methodological contributions in a field. In this spirit, while LiPO pioneered the learning-to-rank framework for preference alignment, our work pioneers a complementary direction by introducing differentiable NDCG optimization for LLM preference alignment.  Both approaches work with ranking response lists rather than pair-wise comparisons and We apologize for causing the misunderstanding that
> > the core contribution of this paper is leveraging the NDCG metric and its continuous proxies to enhance the DPO approach, which originally relies exclusively on pairwise preferences.
>
> Specifically, within this established framework, we advance the methodology by exploring different score functions and leveraging differentiable NDCG optimization to enhance list preference alignment performance. Through extensive experimental validation, we demonstrate the effectiveness of our proposed approach in improving alignment quality.

---

### Official Review · Reviewer_uELb · 2024-11-03

**Soundness:** 3
**Presentation:** 3
**Contribution:** 3
**Rating:** 6
**Confidence:** 3

**Summary:**

1. The paper introduces an approach with differentiable NDCG (diffNDCG) ranking with the aim to improve preference alignment in large language models by optimizing for new ranking-specific metrics. The objective is to improve on simpler optimization objectives that rank items for a recommendation system, e.g. pairwise comparisons of items for a given query, which treat items within a list as independent overlooking the relationship between them.

2. The paper provides a background on traditional preference-alignment models and identifies their limitations to motivate the need for diffNDCG.

3. The diffNDCG approach uses a soft permutation matrix to approximate a ranking order that is differentiable, which allows for gradient-based optimization on the NDCG metric.

4. The differentiable sorting network component (DRPO) offers a way to rank items simultaneously across the entire list based on predicted scores and departs from traditional pairwise comparisons.

**Strengths:**

The authors present a novel approach to preference alignment by introducing differentiable NDCG (diffNDCG), which addresses the challenges of standard ranking models by aligning with the structure of human preference more closely. The paper provides a good motivation for diffNDCG, noting its potential to capture the fact that items in a ranked list are not independent of one another.  The differentiable sorting component, DRPO is what sets it apart from traditional methods.

**Weaknesses:**

1. **Model Misspecification**: The reliance on score-based probabilities could introduce misspecification, as it assumes scores fully capture ranking probabilities through an assumes scoring function.

2. **Biases in Data** Furthermore, if the scoring function is miscalibrated (i.e. depends heavily on what function one is applying to obtain probabilities) or biased (the function weights items in a higher position more), diffNDCG could actually reinforce biases (like position bias), potentially overemphasizing higher-ranked items based on position rather than relevance. Since there's a vast literature on ranking and bias, and this metric's aim is to improve ranking, then this warrants some discussion.

3. **Stand-alone Tables and Experimental Detail**: The tables lack guidance and notes making it difficult to get takeaways without referring back to the text. Tables 2 and 3, for example, do not stand alone. DRPO, ARP, etc. should all be defined in table notes. It is not immediately apparent which metrics indicate improvement or if higher values signify better performance. This should all be in the tables notes. Including confidence intervals and significance tests for key results would also help support the robustness of observed improvements rather than percentage changes.

4. **Experimental Setup and Parameter Transparency**: I'm not clear on the experimental setup. What are the models validated on? Data with labels? Are the clicks simulated? Perhaps this is obvious or self explanatory, but I think should be made clear for any reader.

5. **Writing:* I feel Sections 1-3 could be more to the point, or that we could arrive to the core of the paper that begins us to 3.2 sooner. This would create space for experiments from the Appendix.

The paper would benefit from reducing the amount of passive voice used (rather than: "have demonstrated" "have been proposed" "is developed" "progress has been made" "has been widely adopted") use "they demonstrate" "they propose"). The extensive use of hte passive voice makes it more difficult to follow the paper.

**Questions:**

The authors could address the method’s potential for amplifying biases (position, selection or algorithmic biases), especially if the scoring function is miscalibrated.

Sections 1-3 could be more to the point, or we could arrive to the core of the paper that begins us to 3.2 sooner. This would create space for experiments from the Appendix, including how they set up the validation data.

Tables should be more self-contained with definitions of acronyms and metrics, and notes on takeaways.

The paper would benefit from reducing the amount of passive voice being use (rather than: "have demonstrated" "have been proposed" "is developed" "progress has been made" "has been widely adopted") use "they demonstrated" "they proposed"). It detracts from the clarity of the paper.

**Details Of Ethics Concerns:**

I do not see any ethical concerns with the paper.

---

> ### Author Response · Authors · 2024-11-25
>
> In our future revision, we will enhance the paper's structure by making each section self-contained and eliminating frequent cross-references between Section 3 and later sections. Additionally, we will provide more comprehensive captions in the results Tables for better standalone understanding.
>
> Below are our detailed responses to the specific concerns:
>
>
> **1. Reliability of Ranking Scores in Ranking Preference Datasets Construction.**
>
>
> - In constructing ranking preference datasets, existing list alignment methods [1] have successfully employed score-based winning probabilities as a principled approach for preference labeling. Following this established methodology, we adopt this schema in our work. This schema achieves information-theoretic optimality with robust theoretical guarantees without restrictive assumptions on the underlying probability distributions [2], and has demonstrated strong empirical performance [3] in ranking literature.
>
> - In our understanding, Learning to Rank systems suffer from position bias [4], where users are inherently drawn to items in top positions while paying less attention to lower-positioned items regardless of their actual relevance, leading to multiple forms of bias in the collected datasets. However, in our list alignment task, we employ Large Reward models to score each response, ensuring equal treatment of all responses and thereby reducing position and other potential biases. In this setting, the ranking accuracy primarily depends on the capability of the reward models. Generally, more powerful reward models or advanced language models such as GPT-4 and GPT-4o yield more accurate score evaluations.
>
>
> **2. Experimental Setup and Parameter Transparency**
>
> To verify the effectiveness of our method in aligning large language models with human preferences, we present our experimental setup in Section 4.1 with detailed parameters listed in Appendix E. We evaluate DRPO against various baselines, including SFT, LiPO [1], PRO [5], and DPO [6], across multiple preference datasets [7,8]. To demonstrate scale robustness, we conduct experiments on language models of different scales, ranging from Qwen1.5-0.5B and Qwen1.5-1.8B to Mistral-7B. We will further refine our experimental setup to ensure greater clarity and transparency.
>
> [1] LiPO: Listwise Preference Optimization through Learning-to-Rank.
> [2] Simple, robust and optimal ranking from pairwise comparisons.
> [3] Large language models are effective text rankers with pairwise ranking prompting.
> [4] Mitigate Position Bias with Coupled Ranking Bias on CTR
> [5] Preference Ranking Optimization for Human Alignment.
> [6] Direct Preference Optimization: Your Language Model is Secretly a Reward Model.
> [7] Training a Helpful and Harmless Assistant with Reinforcement Learning from Human Feedback.
> [8] UltraFeedback: Boosting Language Models with Scaled AI Feedback.

---

> > ### Author Response · Authors · 2024-11-25
> >
> > Dear Reviewer uELb,
> >
> > Your profound insights have been tremendously valuable in guiding our work. Understanding your demanding schedule, we deeply appreciate your dedication to reviewing our paper. We have invested considerable effort in addressing the concerns raised during the initial review phase, though we have yet to receive feedback on these revisions.
> >
> > Given that the rebuttal period is drawing to a close, we wanted to respectfully follow up. Your insights on our responses would be invaluable to us at this stage. Should there be any remaining concerns or areas needing further refinement, we stand ready to make additional improvements to our work.
> >
> > We sincerely thank you for your time and constructive feedback throughout this process.

---

### Official Review · Reviewer_D9Sn · 2024-11-05

**Soundness:** 4
**Presentation:** 3
**Contribution:** 4
**Rating:** 6
**Confidence:** 3

**Summary:**

This paper presents a novel approach, Direct Ranking Preference Optimization (DRPO), for aligning Large Language Models (LLMs) with human preferences using ranking preference data. The authors argue that existing methods, primarily focused on pairwise comparisons, fail to adequately capture the nuanced relationships inherent in ranked lists of responses and capture the importance of top ranked responses. DRPO tackles this challenge by framing preference alignment as a Learning-to-Rank (LTR) task and directly optimizing for the Normalized Discounted Cumulative Gain (NDCG) metric, commonly used in LTR.

DRPO consists of three core components:
● Adaptive Rank Policy Score: This component replaces the traditional Policy Reference Ratio score in DPO with a mechanism that prioritizes maximizing the likelihood of preferred responses while incorporating adaptive score margins based on their positions in the ranked list.
● Differentiable Responses Ranking: A differentiable sorting network, specifically an odd-even sorting network, is used to rank responses based on their calculated scores, making the ranking process differentiable.
● DiffNDCG Loss: Due to the non-differentiability of NDCG, the authors propose a differentiable approximation, diffNDCG, which leverages permutation matrices from the sorting network to simulate the NDCG metric.

Finally, extensive experiments on datasets like Anthropic's Helpful Harmless (HH), UltraFeedback, and VLFeedback are used to illustrate that DRPO outperforms baseline methods in metrics like win rates on model based comparisons like GPT-4 and Reward Model Win Rates. The authors also provide an analysis of the time complexity, an ablation study, and computational efficiency of their method.

**Strengths:**

- Novel framing of preference optimization: The paper tackles the crucial problem of LLM preference alignment from a new angle by adopting a listwise ranking approach and optimizing for NDCG, which aligns well with the goal of prioritizing highly preferred responses, and performs very well in practice.

- Technical Soundness: The proposed methodology is technically sound, employing a differentiable sorting network and a differentiable approximation of NDCG to enable end-to-end training.

- Strong Empirical Results: The experimental results are compelling, showing consistent improvements over existing methods across different model sizes and datasets.

- Comprehensive Analysis: The paper includes a thorough ablation study analyzing the impact of individual components, including the Adaptive Rank Policy Score and different discount factors used in diffNDCG. I also appreciate that the reviewers also try existing methods with ARP score and show the positive impact.

**Weaknesses:**

Clarity in Ranking Score Computation: While the concept of Adaptive Rank Policy Score is interesting, the description and motivation behind its specific formulation could be clearer. Providing more intuition and justification for the choices made in Equations (4) and (5) would enhance understanding. The paper goes back and forth between Section 3 and future sections (including results). Authors should reconsider the style of explaining different choices and make the sections self-contained.

- Choice of Sorting Network: The authors opt for the odd-even sorting network due to its "simplicity and ease of implementation". However, they mention other variants with potentially better time complexities. Exploring and comparing different sorting network architectures could strengthen the paper.

- Reliance on availability of scores: DRPO relies on the availability of scalar scores for each response in the ranked list of responses for supervision. This seems to be a limitation when you only have access to ordered pairs (for example, in the Anthropic HH dataset). Does the reliance on DeBERTa in that case change whether the model is able to learn? Could using GPT4 or a larger model make a difference?

- Real-world Human Evaluation: The reliance on reward models as a proxy for human evaluations, while acknowledged as a limitation, could be further addressed. Supplementing the results with some real-world human evaluation of generated responses would bolster the claims. Also, is there a

- Generalizability: The experiments focus on specific datasets. Evaluating DRPO on a wider range of preference alignment tasks and datasets would provide further evidence of its generalizability. See questions below.

**Questions:**

- How sensitive is DRPO to the choice of hyperparameters, particularly those involved in the Adaptive Rank Policy Score and the differentiable sorting network? Policy optimization can be a little unstable so it would be good to present some guide to the reader around the list of hyperparameters and setting them.
- Are there particular scenarios where DRPO is better to optimize for human preference? There are plenty of other preference datasets for LLMs. Can the authors provide a reason for selecting these datasets, and/or the kind of preference datasets where DRPO is expected to work better.
- What are the potential avenues for future research building upon DRPO? For example, could DRPO be adapted for scenarios with dynamic or evolving human preferences?

---

> ### Author Response · Authors · 2024-11-25
>
> In our future revision, we will enhance the paper's structure by making each section self-contained and eliminating frequent cross-references between Section 3 and later sections. Additionally, we will provide more comprehensive captions in the results section for better standalone understanding.
>
> Below are our detailed responses to the specific concerns:
>
> **1. More clear intuition and justification for Adaptive Rank Policy Score**
>
> A common practice in pair-wise alignment methods is to incorporate a margin [1,2] to enhance discrimination between high and low-quality responses. This approach ensures that preferred responses exceed dispreferred ones by at least a specified threshold. To integrate this concept with our list-wise method DRPO, we incorporate a position-dependent term $\gamma(y)$ in our scoring function to account for the score differences computed during response differentiable sorting.
>
> When computing score differences, we obtain a ranking-aware margin $\gamma(y_i)-\gamma(y_j) = \tau \cdot (q(y_i)-q(y_j)) - \beta \cdot ({V}{q(y_i)} - {V}{q(y_j)})$ combines two components: a base margin using relative positions $q(y)$, and a dynamic term controlled by ${V}_{q(y)}$. The base margin follows two principles: higher-ranked elements are more preferred, and adjacent elements have smaller differences.
>
> The dynamic component ${V}_{q(y)}$ adapts to score variations at different ranking positions, expanding the margin for distinct responses and contracting it for finer differentiation when needed."
>
> **2. Network Comparison and Odd-Even Choice**
>
> While both odd-even and bitonic sorting networks have $O(L^2)$ complexity for NDCG computation, their time efficiency is comparable (Table 5). However, DRPO achieves better performance with odd-even sorting, as demonstrated in our experiments below.
>
> |Method|RM Winrate(vs Chosen)|RM Winrate(vs SFT)|
> |---|---|---|
> |DRPO(bitonic)|59.17%($\pm2.78$)|75.58%($\pm2.58$)|
> |DRPO(odd-even)|58.98%($\pm4.95$)|78.90%($\pm3.70$)|
>
>
> **3. Limitations of Reliance on Score Availability**
>
> Utilizing reward models or large language models to compute scalar scores for responses in ranked lists is a widely adopted strategy in practice. For instance, [3] trained a text-to-text reward-ranking model based on T5-XXL to generate scalar scores, while [4] employed large language models for response evaluation. Generally, more powerful reward models or advanced language models such as GPT-4 and GPT-4o yield more accurate score evaluations.
>
>
> **4. Human Evaluation of Generated Responses**
>
> Beyond using reward models as proxies for human evaluations, we employ GPT-4O and GPT-4-turbo as additional evaluators, achieving assessment reliability comparable to human raters. These models have been widely used for response evaluation in several studies [1,3,4,5], and notably, research in [5] demonstrated that GPT-4's assessments achieve agreement rates comparable to inter-human agreement.
>
> **5. Response to Generalizability**
>
> We evaluated DRPO on multiple established datasets beyond HH, including UltraFeedback and VLFeedback - datasets widely adopted in preference alignment research [1,6]. Experiments across various model scales (Qwen1.5-0.5B, Qwen1.5-1.8B, and Mistral-7B) consistently demonstrate DRPO's superior performance over baseline methods.
>
> **6. More Comprehensive Hyperparameter Analysis**
>
> We conducted extensive experiments on hyperparameters used in Adaptive Rank Policy Score and differentiable sorting networks. These experiments demonstrate the robustness of our method. The detailed experimental results are shown below.
>
> - **Experiments On positive constant $\tau$**
>
> |Method|RM Winrate(vs Chosen)|RM Winrate(vs SFT)|
> |---|---|---|
> |DRPO($\tau=0.1$)    |60.50%($\pm2.62$)|77.53%($\pm3.42$)|
> |DRPO($\tau=0.2$)    |58.98%($\pm4.95$)|78.90%($\pm3.70$)|
> |DRPO($\tau=0.5$)    |59.76%($\pm2.73$)|77.92%($\pm2.78$)|
> |DRPO($\tau=1$)      |52.34%($\pm3.70$)|72.07%($\pm3.59$)|
>
> **7. Are there particular scenarios where DRPO is better to optimize for human preference?**
>
> DRPO excels in sophisticated preference alignment tasks, particularly with list-based preferences. We evaluate on HH, UltraFeedback, and VLFeedback datasets, chosen for their comprehensive preference coverage and structured list formats.
>
> **8. What are the potential avenues for future research building upon DRPO? **
>
> DRPO naturally accommodates dynamic human preferences through its list-based structure, making it particularly effective for real-world applications where user preferences evolve over time.
>
> [1] Simpo: Simple preference optimization with a reference-free reward
> [2] Llama 2: Open Foundation and Fine-Tuned Chat Models.
> [3] LiPO: Listwise Preference Optimization through Learning-to-Rank
> [4] TPO: Aligning Large Language Models with Multi-branch & Multi-step Preference Trees
> [5] Direct Preference Optimization: Your Language Model is Secretly a Reward Model
> [6] KTO: Model Alignment as Prospect Theoretic Optimization

---

> > ### Author Response · Authors · 2024-11-25
> >
> > **More Comprehensive Hyperparameter Analysis**
> >
> > We conducted extensive hyperparameter ablation studies, with results detailed below.
> >
> > - **Experiments On update rate $\gamma$**
> >
> > |Method|RM Winrate(vs Chosen)|RM Winrate(vs SFT)|
> > |---|---|---|
> > |DRPO($\gamma=0.99$)    |62.50%($\pm2.92$)|78.12%($\pm2.59$)|
> > |DRPO($\gamma=0.999$)    |59.96%($\pm2.36$)|78.71%($\pm2.61$)|
> > |DRPO($\gamma=0.9999$)    |58.98%($\pm4.95$)|78.90%($\pm3.70$)|
> >
> > - **Experiments on Coefficient $\beta$ in Exponential Moving Average Estimation**
> >
> > |Method|RM Winrate(vs Chosen)|RM Winrate(vs SFT)|
> > |---|---|---|
> > |DRPO($\beta=0.5$)    |57.42%($\pm1.40$)|80.27%($\pm1.50$)|
> > |DRPO($\beta=1$)    |58.98%($\pm4.95$)|78.90%($\pm3.70$)|
> > |DRPO($\beta=2$)    |59.37($\pm4.17$)|80.07%($\pm1.70$)|
> >
> > - **Experiments on Steepness $\alpha$  in Differentiable Sorting Networks**
> >
> > |Method|RM Winrate(vs Chosen)|RM Winrate(vs SFT)|
> > |---|---|---|
> > |DRPO($\alpha=0.1$)    |58.39%($\pm1.77$)|76.36%($\pm2.94$)|
> > |DRPO($\alpha=1$)    |58.98%($\pm4.95$)|78.90%($\pm3.70$)|
> > |DRPO($\alpha=10$)    |59.76%($\pm3.68$)|79.10%($\pm4.95$)|
> > |DRPO($\alpha=50$)    |58.59%($\pm1.46$)|78.71%($\pm0.64$)|

---

> > ### Author Response · Authors · 2024-11-25
> >
> > Dear Reviewer D9Sn,
> >
> > Your profound insights have been tremendously valuable in guiding our work. Understanding your demanding schedule, we deeply appreciate your dedication to reviewing our paper. We have invested considerable effort in addressing the concerns raised during the initial review phase, though we have yet to receive feedback on these revisions.
> >
> > Given that the rebuttal period is drawing to a close, we wanted to respectfully follow up. Your insights on our responses would be invaluable to us at this stage. Should there be any remaining concerns or areas needing further refinement, we stand ready to make additional improvements to our work.
> >
> > We sincerely thank you for your time and constructive feedback throughout this process.

---

> > > ### Comment · Reviewer_D9Sn · 2024-12-02
> > >
> > > Thanks for taking the time to respond to my questions and comments. I really appreciate the new benchmarks and results provided. However, I will keep my score unchanged and will look forward to reading other reviews and discussions.

---

### Meta-Review · Area_Chair_BDrz · 2024-12-24

**Metareview:**

This paper presents Direct Ranking Preference Optimization (DRPO), a novel approach for alignment from human feedback by framing it as a learning-to-rank task optimizing a differentiable approximation of NDCG (diffNDCG. The reviewers recognized the following strengths:
- Novel approach by applying sorting network for differentiable NDCG, and by introducing Adaptive Ranking Policy Score.
-  Experimental results show significant performance gain.

Weaknesses:
- Novelty and contribution compared to prior works are not clear.
- The reason for the significant performance gain should be analyzed and discussed in detail.
- Assuming access to reliable ranking scores seems strong, where in practice the score could be biased or unavailable.
- The contribution of diffNDCG might be overstated.

This is a borderline paper with final ratings of 6, 6, 6, 3. During the discussion, the negative reviewer recognized the performance gain but was not full convinced by the rebuttal. AC agrees with the positive reviews regarding the strengths.  However, AC also agrees with the concerns raised by the negative reviewer such as potential overstatement of contribution and novelty against existing works. The authors are strongly encouraged to carefully revise the final version to address these concerns, which will further enhance the paper. Overall, AC recommends rejection.

**Additional Comments On Reviewer Discussion:**

- Novelty and contribution compared to prior works are not clear. Three reviewers mentioned this weakness, especially concerning novelty against LiPO[1]. The response did not fully convince all reviewers and this is a remaining concern.
 - The reason for the significant performance gain should be analyzed and discussed in detail. The authors explain about possible reasons of performance gain to Reviewer mWpe.
- Assuming access to reliable ranking scores seems strong, where in practice the score could be biased or unavailable. Reviewer uELb and D9Sn shared this concern. The authors argued that in alignment tasks, accessing powerful reward models or directly scoring by powerful LLMs could mitigate the issue. This requirement should be clearly stated in the final version.
- The contribution of diffNDCG might be overstated. AC agrees with Reviewers nU6a that Adaptive Ranking Policy Score (and Eq (5)) is a more interesting and novel idea compared to applying sorting network for approximating NDCG. The authors are suggested to carefully revise the claim about the novelty of diffNDCG to avoid concerns about overstatement.


[1] LiPO: Listwise Preference Optimization through Learning-to-Rank, arXiv:2402.01878.

---

### Decision · Program_Chairs · 2025-01-22

Reject